

# Discrete simulation analysis of COVID-19 and prediction of isolation bed numbers

Xinyu Li[1,2,3,*], Yufeng Cai[2,*], Yinghe Ding[3], Jia-Da Li[2],
Guoqing Huang[4], Ye Liang[1,2] and Linyong Xu[2]

[1] Department of Oral and Maxillofacial Surgery, Center of Stomatology, Xiangya Hospital, Central South University, Changsha, China
[2] Department of Biomedical Informatics, School of Life Sciences, Central South University, Changsha, China
[3] Xiangya School of Medicine, Central South University, Changsha, China
[4] Department of Emergency, Xiangya Hospital, Central South University, Changsha, China
* These authors contributed equally to this work.

Corresponding authors
Ye Liang, liangye@csu.edu.cn
Linyong Xu, xybms@163.com

## ABSTRACT

**Background:** The outbreak of COVID-19 has been defined by the World Health Organization as a pandemic, and containment depends on traditional public health measures. However, the explosive growth of the number of infected cases in a short period of time has caused tremendous pressure on medical systems. Adequate isolation facilities are essential to control outbreaks, so this study aims to quickly estimate the demand and number of isolation beds.

**Methods:** We established a discrete simulation model for epidemiology. By adjusting or fitting necessary epidemic parameters, the effects of the following indicators on the development of the epidemic and the occupation of medical resources were explained: (1) incubation period, (2) response speed and detection capacity of the hospital, (3) disease healing time, and (4) population mobility. Finally, a method for predicting the number of isolation beds was summarized through multiple linear regression. This is a city level model that simulates the epidemic situation from the perspective of population mobility.

**Results:** Through simulation, we show that the incubation period, response speed and detection capacity of the hospital, disease healing time, degree of population mobility, and infectivity of cured patients have different effects on the infectivity, scale, and duration of the epidemic. Among them, (1) incubation period, (2) response speed and detection capacity of the hospital, (3) disease healing time, and (4) population mobility have a significant impact on the demand and number of isolation beds ($P < 0.05$), which agrees with the following regression equation:
$N = P \times (-0.273 + 0.009I + 0.234M + 0.012T1 + 0.015T2) \times (1 + V)$.

## INTRODUCTION

SARS-CoV-2 is a novel coronavirus that has the ability of human-to-human transmission (*Lu et al., 2020*; *Xie & Chen, 2020*). Coronavirus disease 2019 (COVID-19) caused by SARS-CoV-2 has been defined by the World Health Organization (*WHO, 2020a*) as a pandemic (the worldwide spread of a new disease) (*WHO, 2020a*). As of April 5, 2020,

more than 1,100,000 cases of COVID-19 have been reported in different countries and territories (*WHO, 2020b*). Researchers around the world are making every effort to clarify the prevention and control strategy of SARS-CoV-2 (*Chen et al. 2020b*; *Hoffmann et al., 2020*; *Liang et al., 2020*).

COVID-19 is extremely contagious, and its explosive growth in a short space of time has caused tremendous pressure on medical resources (*Bai et al., 2018*). Conventional medical conditions have difficulty meeting the needs of the detection capability for suspected cases and the number of isolation beds for treatment and isolation (*Khan et al., 2020*; *Remuzzi & Remuzzi, 2020*; *Spina et al., 2020*). The number of isolation beds is crucial to reduce the scale of infection and reduce the number of fatalities. Too few isolation beds can lead to the continuation of the epidemic, and too many isolation beds may cause waste and environmental damage (*Bedford et al., 2020*; *Martinez-Alvarez et al., 2020*; *Shah et al., 2020*).

To explore a number of isolation beds, we established a discrete simulation model of epidemics based on COVID-19. By setting several different epidemic indicators we analyzed the changing laws of the epidemic situation, peak value, and scale of the epidemic in different situations. In particular, we pay attention to the occupation of medical resources during the outbreak. We summarized some epidemic indicators related to the number of isolation beds through multiple linear regression and estimated the number of isolation beds through these indicators.

The conclusion is practical, which can provide support for the scheduling of medical resources and the search for effective solutions in the current outbreak or in similar future outbreaks.

## MATERIALS & METHODS

### Establishment of discrete simulation model and parameter interpretation

#### Parameter interpretation

Normal: the health status of the citizens in the city simulation model.

Shadow: too close to patients who have not been isolated, may be infected, and contagious.

Supershadow: different from the general incubation period, it has a longer incubation period and is infectious.

Suspected: refers to the suspected state after the incubation period and is infectious.

Confirmed: refers to the state of being diagnosed with the disease after the hospital's response time. The state has changed from a state of suspected disease and is infectious.

Isolated: refers to the isolated state of being injected into the hospital after the diagnosis, and the bed needs to be occupied.

Cured: refers to the cured state after being diagnosed and can be transferred out of the hospital to vacate the bed.

Dead: refers to the death state of a person who died of illness during the epidemic and no longer has the intention of mobility.

Population mobility rate: the percentage of the population that has willingness to move.

Healing time: the mean time between being in isolation and being discharged.

Incubation period: the time from infection to self-detection of suspected symptoms.

Fatality rate: the probability of death after diagnosis.

Dead time: mean time from diagnosis to death.

Hospital response time: the time from the patient's suspected symptoms to a definitive diagnosis.

Transmission rate: the probability of being infected by contact with an infected person within an unsafe distance.

### Establishment of the discrete simulation model

In this discrete simulation model, we use the Java language to carry out object-oriented programming. The eight states of people were defined as normal, shadow, supershadow, suspected, confirmed, isolated, cured, and dead. At the same time, for the times of being infected, suspected, confirmed, isolated, cured, and died were independent attributes configured to simulate the process with the actual world process, so the program can produce all attributes, including each moment, and record, analyze and calculate the statistics for each simulation individual.

In the model, the length of the incubation period, the time from being a suspected case to being diagnosed, the length of isolation, the rate of population mobility, the probability of infection, and the probability of death after infection can be adjusted or fitted as the necessary parameters for simulation. Most of the time, the parameters follow the normal distribution model, and the mean and standard deviation are defined by the parameters. The probability follows the random number model. In each specific model, the simulation value of the random number model will be given in details. The simulation object state transition logic is shown in Fig. 1.

### General assumptions

a. During the simulation process, the total population of the city is constant, and the number of initial cases is known.

b. The incubation period (shadow time), time of death (died time) and time of cure (cured time) of an individual were in accordance with the normal distribution.

c. There is a fixed response time interval between the onset of patient symptoms and the moment of hospital diagnosis. The hospital always has enough resources to make a diagnosis when the response time is reached.

d. When exposed to an infected person within a dangerous distance, the probability of infection and death after infection is constant.

e. After being cured, the hospital will discharge the patient and be released from isolation at once.

f. Unless otherwise specified, these characteristics do not change over time during disease transmission.

g. The simulation model ends when all patients have been discharged.

h. The patient can not be contagious or infected again after being cured.

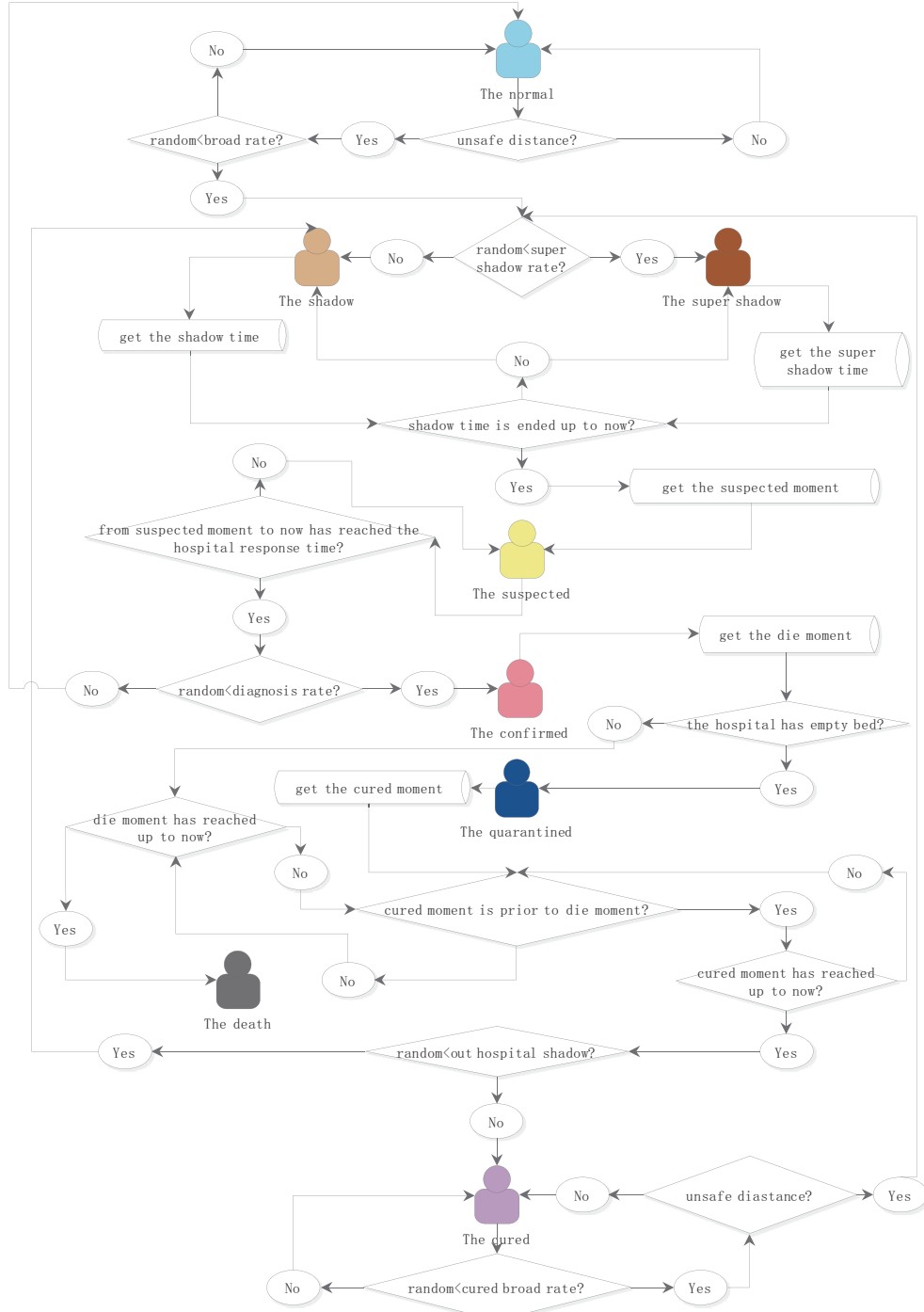

**Figure 1 Logical flowchart of the simulation object status transition.** The eight states of people were defined as normal, shadow, supershadow, suspected, confirmed, isolated, cured, and dead. The times of being infected, suspected, confirmed, isolated, cured, and died were independent attributes configured. The length of the incubation period, the time from being a suspected case to being diagnosed, the length of isolation, the rate of population mobility, the probability of infection, and the probability of death after infection can be adjusted or fitted as the necessary parameters for simulation.

## Model parameters setting

In order to prevent deviations in the simulation process, each set of parameters was repeated 10 times. After removing outliers, the mean was used to draw the curve and analysis.

### Impact of the incubation period on the epidemic development

This model is used to discuss the influence of the disease incubation period on epidemic infection and medical resource occupation. In this model, we make the assumption that the hospital's isolation capacity is strong enough to admit all patients.

To achieve a single variable, we assign the following parameters: the total population = 5,000, number of initially infected persons = 20, population mobility rate = 1, healing time = 15 days (standard deviation = 2 days), hospital response time = 1 day, fatality rate = 0.05, and dead time = 10 days (standard deviation = 5 days) remained the same during the simulation. The experimental groups were as follows: the mean incubation period was set at 3, 7, 10, 14 and 21 days.

### Impact of the hospital response time on the epidemic development

This model is used to discuss the impact of the hospital response time on epidemic infection and medical resource occupation. In this model, the hospital's isolation capacity is strong enough to admit all patients.

To achieve a single variable, we assign the following parameters: the total population = 5,000, number of initially infected persons=20, incubation period = 7 days (standard deviation = 5 days), population mobility rate = 1, healing time = 15 days (standard deviation = 2 days), fatality rate = 0.05, and dead time = 10 days (standard deviation = 5 days) remained the same during the simulation. The experimental groups were as follows: the hospital response time was set at 1, 3, 5, 7 and 10 days.

### Impact of the healing time on the epidemic development

This model is used to discuss the influence of the hospital healing time on the epidemic infection and medical resource occupation. In this model, the hospital's isolation capacity is strong enough to admit all patients.

To achieve a single variable, we assign the following parameters: the total population = 5,000, number of initially infected persons = 20, incubation period = 7 days (standard deviation = 5 days), population mobility rate = 1, hospital response time = 1 day, fatality rate = 0.05, and dead time = 10 days (standard deviation = 5 days) remained the same during the simulation. The experimental groups were as follows: the hospital healing time was set at 5, 7, 10, 15 and 20 days

### Impact of the population mobility on the epidemic development

This model was used to discuss the influence of the population mobility on the epidemic infection and medical resource occupation. In this model, the hospital's isolation capacity is strong enough to admit all patients.

To achieve a single variable, we assign the following parameters: the total population = 5,000, number of initially infected persons = 20, incubation period = 7 days (standard deviation = 5 days), hospital response time = 1 day, healing time = 15 days

(standard deviation = 2 days), fatality rate = 0.05, and dead time = 10 days (standard deviation = 5 days) remained the same during the simulation. The experimental groups were as follows: the population flow rate was set to 0, 30%, 50%, 80% and 100%.

### Impact of the hospital isolation capacity on the infection situation

This model was used to discuss the influence of the hospital isolation capacity on the epidemic infection.

To achieve a single variable, we assign the following parameters: the total population = 5,000, number of initially infected persons = 20, incubation period = 7 days (standard deviation = 5 days), population mobility rate = 1, hospital response time = 1 day, healing time = 15 days (standard deviation = 2 days), fatality rate = 0.05, and dead time = 10 days (standard deviation = 5 days) remained the same during the simulation. The experimental groups were as follows: the hospital isolation capacity was set to 40%, 60%, 80% and 100%.

## Multiple regression analysis and parameter prediction of the number of isolation facilities

Multilinear regression (MLR) analysis was used to evaluate the impact of the simulation parameters on the dependent variable demand for the number of isolation beds. Meanwhile, the prediction method was applied to an example under different healing times to test the accuracy.

## Statistical analysis

Continuous variables are compared using the average. Using SPSS v26.0 (IBM authorized Central South University to use) for data analysis, analysis of variance (ANOVA) was used to analyze the level of significant difference between the groups. When the variance is homogeneous, the least significant difference (LSD) method and multiple comparisons are used to analyze between any two groups; when the variance is not homogeneous, Tamhane's T2 tests and the multiple comparison are used to compare the mean between any two groups. The Grubbs method was used to address outliers. For all statistical analyses, the test level was $\alpha = 0.05$.

R is used to represent the goodness of fit of the multiple regression to measure the fitting degree of the estimated model to the observed values. The table of regression analysis lists the results of the significance test of independent variables (using t-test) and the $P$ value of t-test, indicating whether independent variables have a significant influence on dependent variables.

## RESULTS

### Impact of epidemic factors on the infection situation under sufficient isolation facilities

#### Impact of incubation period on the epidemic development

The incubation period is an asymptomatic stage in the early stages of disease development, at which point patients themselves will not suspect that they have been infected.

**Table 1 Epidemic indicators under different incubation periods.**

| Group | Incubation period | Maximum newly confirmed cases | | Maximum cases in incubation period | | Maximum daily effective reproductive number | Sum of infected cases | Duration of the epidemic | Maximum inpatient cases | |
|---|---|---|---|---|---|---|---|---|---|---|
| | | Number | Date | Number | Date | | | | Number | Date |
| 1 | 3 | 74[bcde] | 19[bcde] | 271[bcde] | 17[bcde] | 1.14[bcde] | 1,572[bcde] | 87[bcde] | 786[bcde] | 26[bcde] |
| 2 | 7 | 113[ade] | 24[acde] | 774[acde] | 20[ae] | 1.50[ade] | 2,660[acde] | 114[ade] | 1,297[acde] | 31[acde] |
| 3 | 10 | 128[ade] | 27[abe] | 1192[abde] | 21[ae] | 1.78[ade] | 3,146[abde] | 135[ad] | 1,475[abde] | 34[abde] |
| 4 | 14 | 147[abc] | 30[abe] | 1790[abce] | 22[ae] | 2.30[abce] | 3,518[abce] | 164[ab] | 1,675[abc] | 37[abce] |
| 5 | 21 | 148[abc] | 36[abcd] | 2436[abcd] | 26[abcd] | 3.82[abcd] | 3,901[abcd] | 202[abc] | 1,722[abc] | 45[abcd] |

Notes:
[a] There is a significant difference compared with group 1 ($p < 0.05$).
[b] There is a significant difference compared with group 2 ($p < 0.05$).
[c] There is a significant difference compared with group 3 ($p < 0.05$).
[d] There is a significant difference compared with group 4 ($p < 0.05$).
[e] There is a significant difference compared with group 5 ($p < 0.05$).

We compared the infection situation of different incubation periods. ANOVA showed that the mean values of all indicators among groups were not exactly the same ($P < 0.05$). The detailed differences among groups are shown in Table 1.

Further analysis was performed using multiple comparisons. The maximum number of incubation cases, the sum of infected cases and the corresponding date for the peak number of inpatients were significantly different between any two groups ($P < 0.05$). The long incubation period promoted these epidemic indicators. We use the Basic Reproduction Number (R0) to describe the rate of infection of an epidemic. This value can reflect the possibility and severity of an infectious disease outbreak. Basic Reproduction Number (R0) refers to the number of people who can be infected by an average patient in an environment with all susceptible people without intervention. For common epidemics, R0 will be bigger than one. In addition, Effective Reproduction Number (Rt) represents the average number of patients that can be infected at a certain moment in the process of disease transmission and development, with the addition of prevention and control interventions, such as isolation of patients in shelters, isolation of individuals at home, and wearing masks. The maximum number of newly confirmed cases and their corresponding dates, corresponding date of peak incubation cases, maximum value of Rt, duration of the epidemic, and maximum number of inpatients were not exactly equal among the groups in different incubation periods, and there were significant differences among some groups ($P < 0.05$). The above indicators increased with the increase in the incubation period (Fig. 2).

### Impact of the hospital response time on the epidemic development

In the simulation model, hospital response time refers to the time it takes for a patient to develop a first symptom until a clear diagnosis is obtained. This time is extended if the patient is unwilling to go to the hospital to undergo detection testing, if the hospital detection method fails to diagnose the disease early, or if there are insufficient hospital detection reagents.

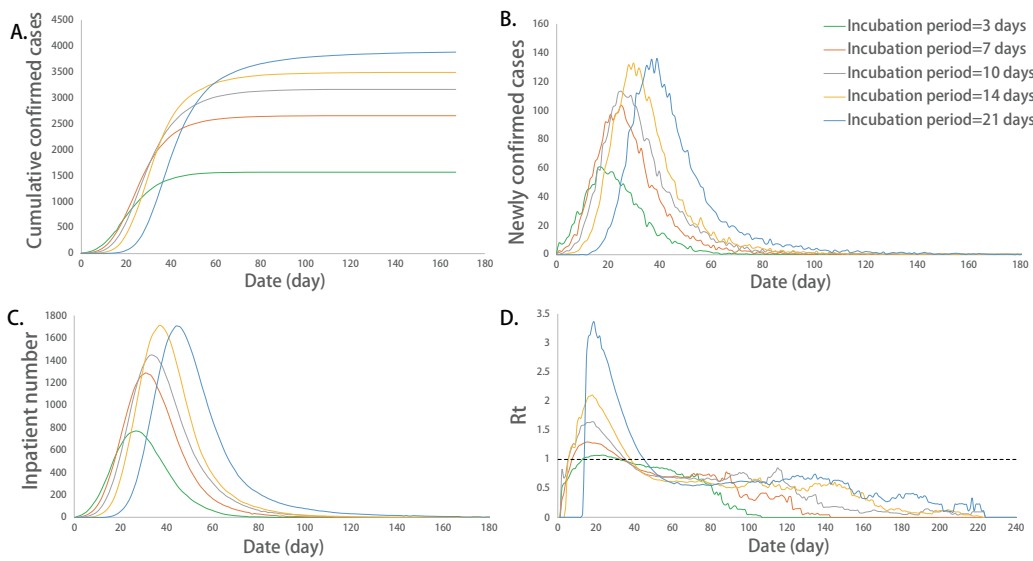

**Figure 2 Epidemic development curve under different incubation periods.** (A) The time-varying curves of the cumulative number of confirmed cases. (B) The time-varying curves of the cumulative number of the number of newly confirmed cases. (C) The time-varying curves of the cumulative number of the number of inpatient cases. (D) The time-varying curves of the cumulative number of the daily effective reproductive number. The data for each curve is the average of 10 simulations. The epidemic peak values increased and their corresponding dates delayed with the increase in the incubation period.

**Table 2 Epidemic indicators under different hospital response times.**

| Group | Hospital response time | Maximum newly confirmed cases | | Maximum cases in incubation period | | Maximum daily effective reproductive number | Sum of infected cases | Duration of the epidemic | Maximum inpatient cases | |
|---|---|---|---|---|---|---|---|---|---|---|
| | | Number | Date | Number | Date | | | | Number | Date |
| 1 | 1 | 113[bcde] | 24[cde] | 774[bcde] | 20 | 1.50[bcde] | 2,660[bcde] | 114[ce] | 1,297[bcde] | 31[cde] |
| 2 | 3 | 131[ade] | 25[cde] | 935[ae] | 19 | 2.06[acde] | 2,985[acde] | 134[e] | 1,508[ade] | 32[de] |
| 3 | 5 | 133[ade] | 27[ae] | 945[ae] | 19 | 2.80[abe] | 3,213[abde] | 155[a] | 1,555[ade] | 35[ae] |
| 4 | 7 | 146[abc] | 29[abce] | 1,010[ae] | 18 | 3.27[abe] | 3,469[abce] | 136[e] | 1,701[abc] | 36[abe] |
| 5 | 10 | 155[abc] | 32[abcd] | 1,110[abcd] | 19 | 4.98[abcd] | 3,695[abcd] | 181[abd] | 1,802[abc] | 39[abcd] |

**Notes:**
[a] There is a significant difference compared with group 1 ($p < 0.05$).
[b] There is a significant difference compared with group 2 ($p < 0.05$).
[c] There is a significant difference compared with group 3 ($p < 0.05$).
[d] There is a significant difference compared with group 4 ($p < 0.05$).
[e] There is a significant difference compared with group 5 ($p < 0.05$).

ANOVA showed that there were no significant differences in the corresponding date of the maximum number of incubation cases among different response time groups ($P > 0.05$), and the mean of the other indicators among the groups was not exactly equal ($P < 0.05$). The detailed differences among the groups are shown in Table 2.

The multiple comparisons showed that the sum of infected cases between any two groups was significantly different ($P < 0.05$). The sum of the infected cases increased with the extension of response time. The maximum number of newly confirmed cases and its corresponding date, the maximum number of incubation cases, the maximum of Rt, the

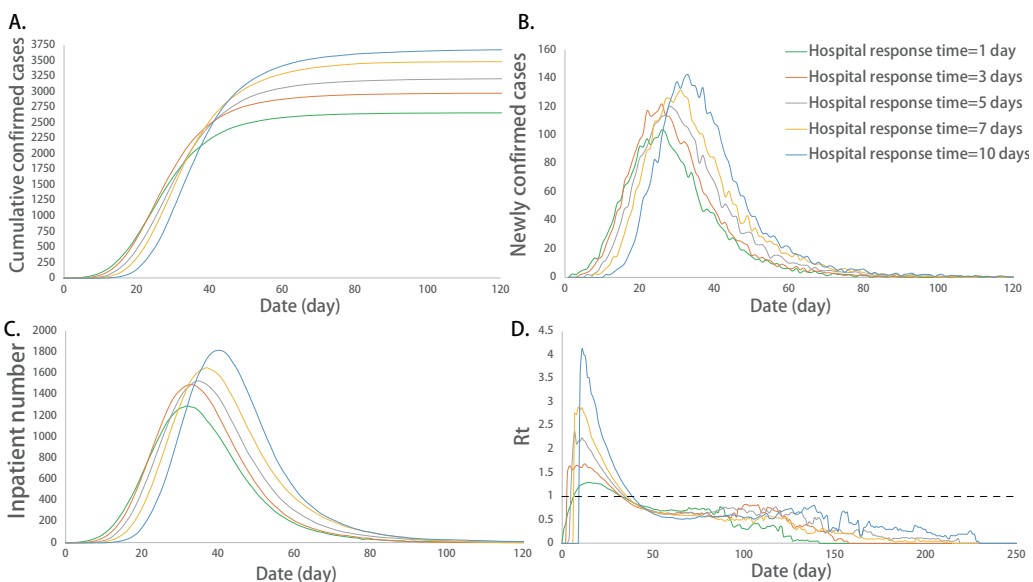

**Figure 3 Epidemic development curve under different hospital response time.** (A) The time-varying curves of the cumulative number of confirmed cases. (B) The time-varying curves of the cumulative number of the number of newly confirmed cases. (C) The time-varying curves of the cumulative number of the number of inpatient cases. (D) The time-varying curves of the cumulative number of the daily effective reproductive number. The data for each curve is the average of 10 simulations. The hospital's slow response speed caused an increase in the epidemic scale and a delay in the peak of the outbreak.

duration of the epidemic, the maximum number of inpatients and its corresponding date were not exactly equal among the groups at different response times, and there were significant differences among some groups ($P < 0.05$). The above indicators increased with the extension of the response time (Fig. 3).

### Impact of the healing time on the epidemic development

The healing time refers to the average time from admission to discharge. ANOVA showed that the mean values of the maximum value of Rt, the maximum number of inpatients and their corresponding dates among different healing time groups were not exactly the same ($P < 0.05$), and the other indicators were not considered to be significantly different ($P > 0.05$). The detailed differences among the groups are shown in Table 3.

Further multiple comparisons showed that the maximum number of inpatients between any two groups was significantly different ($P < 0.05$). The maximum number of inpatients increased with the extension of the healing time. The maximum value of Rt and the corresponding date of peak inpatient number were not exactly equal among the groups at different healing times, and there were significant differences among some groups ($P < 0.05$). The extension of the healing period promoted the increase in the above indicators (Fig. 4).

### Impact of the population mobility on the epidemic development

The population mobility rate refers to the proportion of people in motion to the total population. We compared the infection situation of different population mobility rates.

**Table 3 Epidemic indicators under different healing times.**

| Group | Healing time | Maximum newly confirmed cases | | Maximum cases in incubation period | | Maximum daily effective reproductive number | Sum of infected cases | Duration of the epidemic | Maximum inpatient cases | |
|---|---|---|---|---|---|---|---|---|---|---|
| | | Number | Date | Number | Date | | | | Number | Date |
| 1 | 5 | 114 | 22 | 798 | 19 | 1.86[cde] | 2,677 | 95 | 493[bcde] | 24[cde] |
| 2 | 7 | 117 | 24 | 834 | 18 | 1.74[ce] | 2,603 | 113 | 694[acde] | 26[ce] |
| 3 | 10 | 111 | 23 | 792 | 19 | 1.38[ab] | 2,591 | 104 | 942[abde] | 28[ab] |
| 4 | 15 | 113 | 24 | 774 | 20 | 1.50[a] | 2,660 | 114 | 1,297[abce] | 31[a] |
| 5 | 20 | 112 | 23 | 777 | 18 | 1.36[ab] | 2,643 | 108 | 1,590[abcd] | 33[ab] |

Notes:
[a] There is a significant difference compared with group 1 ($p < 0.05$).
[b] There is a significant difference compared with group 2 ($p < 0.05$).
[c] There is a significant difference compared with group 3 ($p < 0.05$).
[d] There is a significant difference compared with group 4 ($p < 0.05$).
[e] There is a significant difference compared with group 5 ($p < 0.05$).

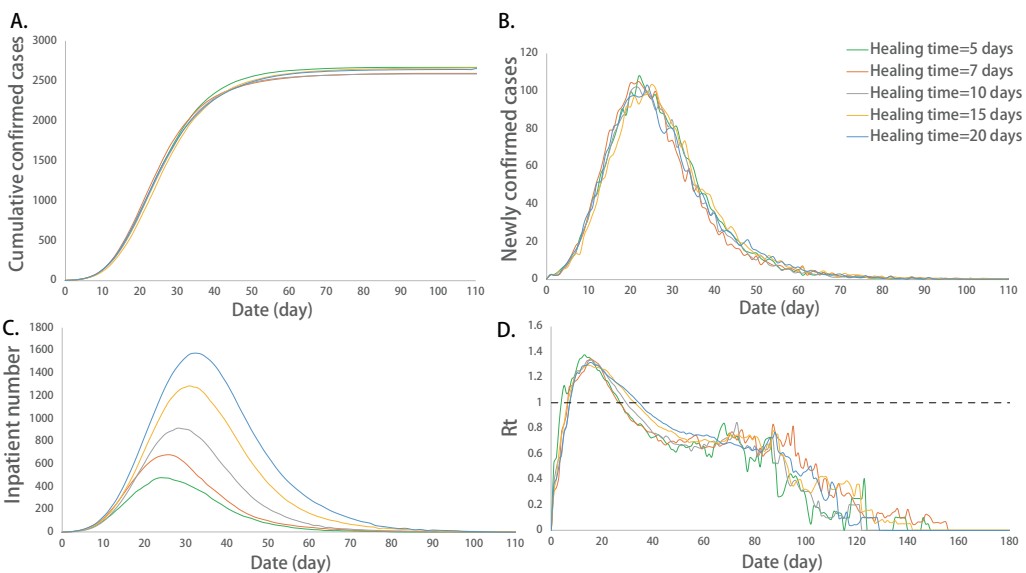

**Figure 4 Epidemic development curve under different healing time.** (A) The time-varying curves of the cumulative number of confirmed cases. (B) The time-varying curves of the cumulative number of the number of newly confirmed cases. (C) The time-varying curves of the cumulative number of the number of inpatient cases. (D) The time-varying curves of the cumulative number of the daily effective reproductive number. The data for each curve is the average of 10 simulations. In this model, the ability of hospital treatment had no effect on the number of cumulative and newly confirmed cases. But the maximum number of inpatients increased and its corresponding peak dates delayed with the extension of the healing time.

ANOVA showed that there were no significant differences in the maximum value of Rt and the duration of the epidemic among the groups ($P > 0.05$), and the mean of other indicators among the groups was not exactly equal ($P < 0.05$). The detailed differences among the groups are shown in Table 4.

**Table 4 Epidemic indicators under different population mobility rates.**

| Group | Population mobility rate | Maximum newly confirmed cases | | Maximum cases in incubation period | | Maximum daily effective reproductive number | Sum of infected cases | Duration of the epidemic | Maximum inpatient cases | |
|---|---|---|---|---|---|---|---|---|---|---|
| | | Number | Date | Number | Date | | | | Number | Date |
| 1 | 100% | 113[cde] | 24 | 774[cde] | 20[e] | 1.50 | 2,660[bcde] | 114 | 1,297[cde] | 31[e] |
| 2 | 80% | 98[cde] | 23 | 681[cde] | 20[e] | 1.40 | 2,477[acde] | 117 | 1,165[cde] | 31 |
| 3 | 50% | 70[abde] | 25 | 456[abde] | 22[e] | 1.33 | 1,994[abde] | 119 | 799[abde] | 34[e] |
| 4 | 30% | 34[abce] | 26 | 203[abce] | 23[e] | 1.55 | 1,204[abce] | 133 | 368[abce] | 35[e] |
| 5 | 0% | 9[abcd] | 17 | 43[abcd] | 11[abcd] | 1.26 | 209[abcd] | 108 | 74[abcd] | 24[acd] |

Notes:
[a] There is a significant difference compared with group 1 ($p < 0.05$).
[b] There is a significant difference compared with group 2 ($p < 0.05$).
[c] There is a significant difference compared with group 3 ($p < 0.05$).
[d] There is a significant difference compared with group 4 ($p < 0.05$).
[e] There is a significant difference compared with group 5 ($p < 0.05$).

The multiple comparisons showed that the sum of the infected cases between any two groups was significantly different ($P < 0.05$). The increase in the population mobility rate caused a higher sum of the infected cases. The maximum number of newly confirmed cases and its corresponding date, the maximum number of incubation cases and its corresponding date, the maximum number of inpatients and their corresponding dates were not exactly equal among the groups in different population mobility rates, and there were significant differences among some groups ($P < 0.05$). Among them, the maximum number of newly confirmed cases, the maximum number of incubation cases and the maximum number of inpatients increased with the increase in the population activity rate. At the extreme value of 0%, that is, when everyone was inactive, the corresponding date of peak incubation cases and the corresponding date of peak inpatient number were significantly advanced, which was significantly different from that of the other groups (Fig. 5).

## Impact of the hospital isolation capacity on the infection situation

The hospital isolation capacity is defined by the proportion of the actual quantity of isolation beds to the demanded quantity for isolation beds. ANOVA showed that there were no significant differences in the maximum number of newly confirmed cases, the maximum number of incubation cases and the maximum value of Rt among different isolation capacity groups ($P > 0.05$), and the mean of the other indicators among the groups was not exactly equal ($P < 0.05$). The detailed differences among groups are shown in Table 5.

The multiple comparisons showed that the corresponding dates of the peak number of inpatients and the duration of isolation facilities at their full capacity between any two groups were significantly different ($P < 0.05$). The corresponding date of the peak inpatient number was delayed with the decrease in isolation capacity. The duration of isolation facilities at their full capacity increased with the lack of isolation beds, which showed a quadratic relationship (Table S1, Fig. S1). In addition, the corresponding date of peak newly confirmed cases, the corresponding date of incubation cases, the sum of infected

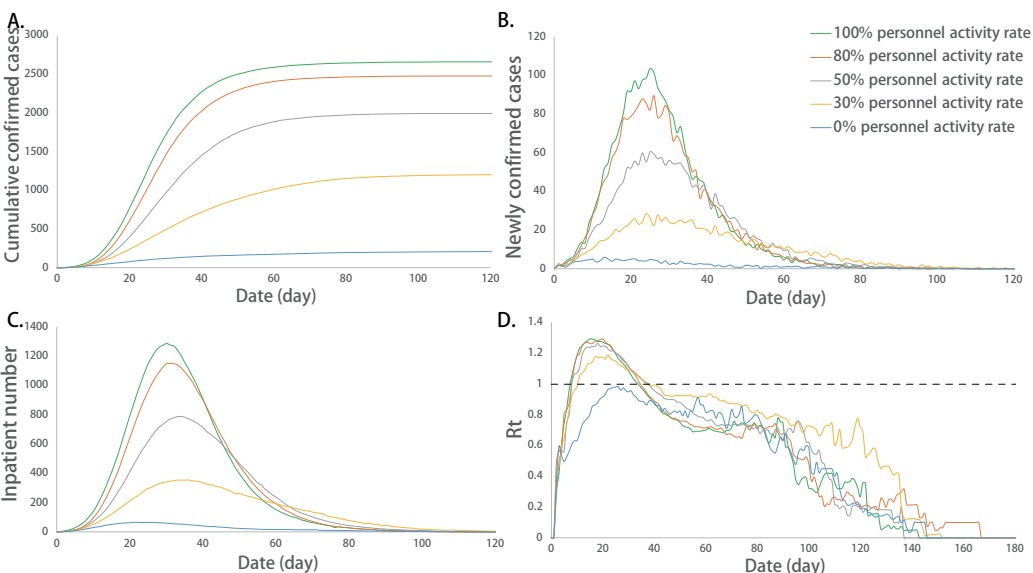

**Figure 5 Epidemic development curve under different population mobility rates.** (A) The time-varying curves of the cumulative number of confirmed cases. (B) The time-varying curves of the cumulative number of the number of newly confirmed cases. (C) The time-varying curves of the cumulative number of the number of inpatient cases. (D) The time-varying curves of the cumulative number of the daily effective reproductive number. The data for each curve is the average of 10 simulations. Restrictions on population mobility made the peak number of cumulative confirmed cases, newly confirmed cases, and inpatient cases decreased. Population mobility rate had no effect on the value of Rt. However, the different transmission laws are displayed in extreme cases.

**Table 5 Epidemic indicators under different hospital isolation capacities.**

| Group | Hospital isolation capacity | Maximum newly confirmed cases | | Maximum cases in incubation period | | Maximum daily effective reproductive number | Sum of infected cases | Duration of the epidemic | Duration of isolation facilities at full load | Peak admission date |
|---|---|---|---|---|---|---|---|---|---|---|
| | | Number | Date | Number | Date | | | | | |
| 1 | 40% | 119 | 25[c] | 793 | 23[bcd] | 1.40 | 4,089[bcd] | 143[bcd] | 82[bcd] | 18[bcd] |
| 2 | 60% | 115 | 24[c] | 794 | 19[a] | 1.34 | 3,275[acd] | 103[a] | 34[acd] | 22[acd] |
| 3 | 80% | 114 | 22[ab] | 802 | 18[a] | 1.37 | 2,769[ab] | 105[a] | 13[abd] | 25[abd] |
| 4 | 100% | 113 | 24 | 774 | 20[a] | 1.50 | 2,660[ab] | 114[a] | 0[abc] | 31[abc] |

Notes:
[a] There is a significant difference compared with group 1 ($p < 0.05$).
[b] There is a significant difference compared with group 2 ($p < 0.05$).
[c] There is a significant difference compared with group 3 ($p < 0.05$).
[d] There is a significant difference compared with group 4 ($p < 0.05$).

cases and the duration of the epidemic were not exactly equal among the groups, and there were significant differences among some groups ($P < 0.05$). In the severely inadequate isolation capacity group (40%), the corresponding date of incubation cases, the mean value of Rt, the sum of infected cases and the duration of the epidemic were significantly increased ($P < 0.05$) (Fig. 6).

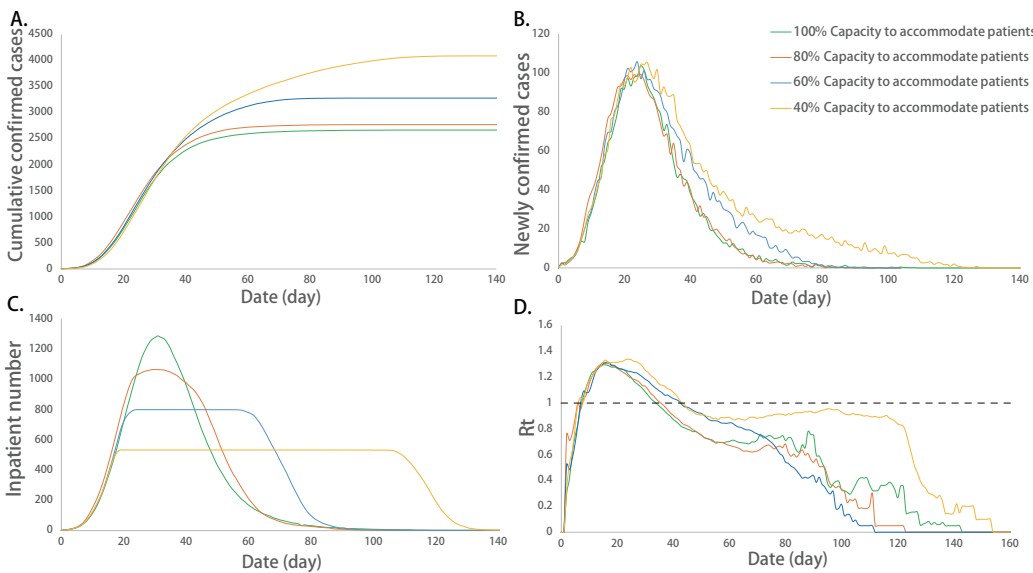

**Figure 6 Epidemic development curve under different hospital isolation capacities.** (A) The time-varying curves of the cumulative number of confirmed cases. (B) The time-varying curves of the cumulative number of the number of newly confirmed cases. (C) The time-varying curves of the cumulative number of the number of inpatient cases. (D) The time-varying curves of the cumulative number of the daily effective reproductive number. The data for each curve is the average of 10 simulations. The peak number of cumulative confirmed cases and inpatient cases increased with the decrease in the isolation capacity. In addition, with the decrease in isolation capacity, the corresponding date of the peak inpatient number was delayed and the duration of isolation facilities at their full capacity increased. Isolation capacity had no effect on the newly confirmed cases and the value of Rt.

## Multiple regression analysis and parameter prediction of the number of isolation facilities

### Multivariate regression analysis

To further explore the rational setting of isolation beds in the medical system under multifactor epidemic conditions, we analyzed the relationship between different epidemic indicators and a number of isolation beds by multiple regression analysis. The t-test showed that the independent variables of incubation period, population mobility rate, hospital response time and healing time significantly affected the number of isolation beds ($P < 0.05$) (Table 6). Finally, we obtained the following regression equation ($R^2 = 0.841$):

$$N = P \times (-0.273 + 0.009I + 0.234M + 0.012T1 + 0.015T2) \times (1 + V)$$

$N$ indicates a number of isolation beds. $P$ represents population, which refers to the total population in the corresponding area. I represents incubation period of the epidemic. M represents population mobility rate, which refers to the proportion of people in motion to the total population. T1 represents hospital response time, which refers to the time it takes for a patient to develop the first symptom until a clear diagnosis is obtained. T2 represents healing time, which refers to the average time from admission to discharge. T1 and T2 can be estimated based on a certain number of cases. Considering that a certain number of isolation beds should be reserved to cope with emergency situations,
**Table 6 Multiple-linear regression analysis of the reasonable number of isolation beds.** The table lists the results of the *t*-test and its *P*-value, indicating whether independent variables have a significant influence on dependent variables. The standard expression coefficient reflects the influence degree of each independent variable on the dependent variable after eliminating the influence of the dependent variable and the unit taken by the independent variable.

| Independent variables | Regression coefficient | Standard regression coefficient | t | P |
|---|---|---|---|---|
| Constant | −0.273 | | −15.226 | 0.000** |
| Incubation period | 0.009 | 0.302 | 10.276 | 0.000** |
| Population mobility rate | 0.234 | 0.629 | 21.046 | 0.000** |
| Hospital response time | 0.012 | 0.274 | 9.178 | 0.000** |
| Healing time | 0.015 | 0.452 | 15.376 | 0.000** |

**Note:**
** $p < 0.01$.

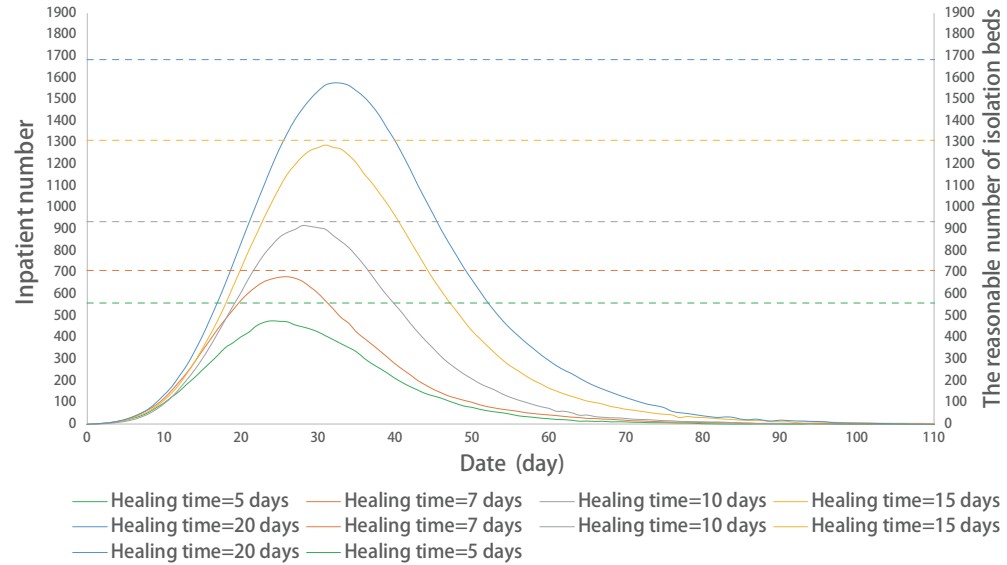

**Figure 7 Prediction of a reasonable number of isolation beds in different healing times.** The solid line represents the time-varying curves of the number of inpatient cases at different healing times; the dotted line represents the reasonable number of isolation beds at different healing times.

we set V as the reserve amount. This article suggests that it is generally reserved at 10% according to the simulation results.

### Prediction of a number of isolation beds in different healing times

Here, the prediction method was applied to an example (Fig. 7). The predicted number of isolation beds were basically consistent with the model operating results. The parameter setting is the same as model 2.3.

## DISCUSSION

### Impact of the incubation period on the epidemic development

According to result 1.1, we found that a longer incubation period significantly promoted the infectivity, scale and duration of the epidemic. By tracking this phenomenon, it was

found that the patients in the incubation period and without mobility restriction caused a high level of transmission before displaying symptoms by contacting others, which was similar to the views of *Jiang, Rayner & Luo (2020)* and *Li et al. (2020)*. To contain an outbreak, early detection of suspected cases is critical (*De Salazar et al., 2020*). Some studies have described that a longer incubation period may be beneficial for epidemic control (*Kahn et al., 2020*), as this allows the Centers for Disease Control and Prevention (CDC) to have more time to deal with the overall epidemic. This conclusion may be more applicable to some known diseases, but for unknown diseases, we believe that a longer incubation period represents a more dangerous signal, making the development of the epidemic uncontrollable (*Kahn et al., 2020*; *Lauer et al., 2020*). Due to the reduced predictability of the disease outbreak scale, it is more difficult to track patients. As a result, the disease may spread to a wider range of people, making it difficult to control.

Our results showed that increasing hospital response speed could improve infectivity and scale of an outbreak, which is consistent with previous research (*Chan et al., 2020*; *Hellewell et al., 2020*; *Wu, Leung & Leung, 2020*). Shortening the hospital response time depends on the public's awareness of epidemic prevention and the level of medical technology. On the one hand, the public needs to pay more attention to the epidemic and actively cooperate with early detection; on the other hand, medical technology determines the time it takes for the detection method to give an accurate result. In addition, if sufficient isolation facilities can be provided, the centralized isolation of all suspected patients who cannot be excluded can also help reduce the hospital response time (*Bai et al., 2018*; *Chen et al. 2020a*).

Compared with the hospital response time, the impact of the population mobility rate on the duration of the epidemic is not significant in the case of abundant medical resources. However, unrestricted population mobility can cause a large medical load and consumption of medical supplies and will generate a large number of infected cases, resulting in adverse socioeconomic impacts (*Kraemer et al., 2020*; *Tian et al., 2020*; *Yang et al., 2020*). In the actual epidemic situation, the amount of medical resources will be less and more scarce than usual. Therefore, it cannot be considered that the population mobility has no effect on the duration of the outbreak. The detailed consequences of inadequate medical resources are discussed in section 3.2. In extreme cases, all people stopped activities, i.e., when the mobility rate is 0%, the overall scale of the epidemic is dramatically reduced, and the corresponding date of the epidemic peak sharply advances. It is speculated that the disease may saturate in a small area after taking this extreme prevention and control measure; thus, the transmission will be completely blocked. Since there is no new-generation infection, different transmission laws are displayed.

## Impact of insufficient isolation facilities on the epidemic development

There is no specific treatment or vaccine for COVID-19 at the time of the conceptualization of the simulation model, containment of the epidemic depends more on traditional public health measures (*Wilder-Smith & Freedman, 2020*). The results showed that the low capacity of hospital isolation would lead to serious consequences. This is consistent with previous studies showing that intervention through timely isolation

measures is effective in controlling the epidemic (*Fraser et al., 2004*; *Rocklov, Sjodin & Wilder-Smith, 2020*). Even with vaccines, adequate isolation beds are a key factor in controlling the outbreak. In our model, if you need to simulate the possession of a vaccine, you can simply modify the total population. The method is to update this constant to subtract the number of people immunized.

In addition, the simulation model showed that the duration of isolation facilities at full capacity showed a quadratic relationship with the gap in the number of isolation facilities. That is, as the gap in the number of isolated beds expands, the growth rate of full-load time gradually accelerates, which may increase the uncontrollability of the epidemic. In our model, the hospital isolation capacity does not affect the ability of to detect the disease and diagnosis patients, so it has no significant effect on the maximum number of newly confirmed cases and their corresponding dates. However, the increase in the duration of isolation facilities at full load shows a more severe overload of the medical system. Thus, under the condition of insufficient isolation capacity, expanding the original size of the hospital or constructing Fangcang shelter hospitals are the key measures to contain outbreaks (*Chen et al. 2020a*; *Khan et al., 2020*; *Wu & McGoogan, 2020*).

## Multiple regression analysis and parameter prediction of the number of isolation facilities

We summarized the related factors that affected the demanded quantity of isolation beds by multiple regression analysis. Furthermore, multivariate regression analysis was used to estimate the number of isolation beds:

$$N = P \times (-0.273 + 0.009I + 0.234M + 0.012T1 + 0.015T2) \times (1 + V)$$

The regression equation shows that the population mobility rate is the variable with the highest weight, which indicates that the restriction of population mobility is the critical factor to contain outbreaks and effectively reduce the load on the medical system (*Liao et al., 2020*; *Ng et al., 2020*; *Tian et al., 2020*; *Wu, Leung & Leung, 2020*). We believe that reducing the epidemic scale by restricting population mobility can also help to provide time for the establishment of temporary isolation.

In practice, the incubation period (I) can be estimated from the time between traceable harmful exposure to the time of the first symptom (*Backer, Klinkenberg & Wallinga, 2020*; *Lauer et al., 2020*). The population mobility rate (M) can be roughly estimated by the ratio of the population in unrestricted mobility, including medical personnel and administrative personnel, to the total population. (*Telionis et al., 2020*; *Wu, Leung & Leung, 2020*). Hospital response time (T1) and healing time (T2) can be estimated based on a certain number of cases. In addition, we recommend V = 10% as a reserve to address emergency situations under actual conditions. The above indicators are easy to obtain and estimate, which provides a feasible guarantee for using this method to estimate a number of isolation facilities.

More importantly, estimating the number of isolation facilities based on the epidemic situation and relevant parameters of the medical system will help to predict the pressure of the medical system in different areas in advance. This will provide decision-making support for the rational arrangement of medical resources and epidemic control.

### Advantages and limitations of this study

Based on the data we sampled, the simulation model proposed in the manuscript can analyze the common parameters of infectious diseases and the related laws of transmission in the case of a single variable. However, we must admit that this model is still preliminary. As researchers of this issue, we have summarized the advantages and limitations of the study.

Advantage: (1) The established simulation program is controlled by computer advanced language software, using multi-threaded parallel operation methods. It can produce results quickly. (2) The simulation program is convenient, and the modification parameters can be subsequently analyzed for different types of epidemic. The output value is a continuous variable, which is good for statistical analysis. (3) Parametric assumptions based on data analysis are established in a virtual environment, which is conducive to early warning of risks.

Limitation: (1) This study attempts to put forward a more general model to simulate the process of disease transmission. But in fact, we only tested on part of the COVID-19 data. If the model is removed from this environment, it may produce large errors. (2) In this model, we consider many common parameters, most of which we give suggestions for interpretation and value in the paper. If we want to pursue higher accuracy of the model, we strongly suggest that the model should be verified by the local actual situation in the early stage.

# CONCLUSIONS

The incubation period, response speed detection capacity of the hospital, disease healing time, and degree of population mobility have different effects on the infectivity, scale, duration of the epidemic, and the demand and number of isolation beds. Adequate isolation facilities are essential to control outbreaks. The prediction equation can be easily and quickly applied to estimate the demand number of isolation beds in a COVID-19-affected city. This study will provide decision-making support for the rational arrangement of medical resources and epidemic control.

# ACKNOWLEDGEMENTS

We thank Bruce Yong for providing a prototype of the program. We referred to his open-source virus broadcast simulation project and used it as the prototype of our simulation design. With Bruce Yong's consent, we redesigned some programs according to the epidemic situation. Thank you to Prof. Zhoushun Zheng, Prof. Canhua Jiang and Prof. Changyun Fang of Central South University for their support for this project. Thank you to the MAIGO mathematical modeling team for their help with this project.

### Funding

This work was supported by the National Natural Science Foundation of China (Grant No. 81901065), the Hunan Province Science, Technology Department (Hunan Natural Science

Fund - Youth Foundation Project, 2018JJ3850), the Hunan Health Commission (B2019192), the Natural Science Foundation of Hunan province (2017JJ2392), the Scientific research project of Hunan provincial health commission (B20180054), and the Changsha Science and Technology project (kq1706072). The funders had no role in study design, data collection and analysis, decision to publish, or preparation of the manuscript.

### Grant Disclosures
The following grant information was disclosed by the authors:
National Natural Science Foundation of China: 81901065.
Hunan Province Science, Technology Department (Hunan Natural Science Fund—Youth Foundation Project): 2018JJ3850.
Hunan Health Commission: B2019192.
Hunan Province: 2017JJ2392.
Hunan Provincial Health Commission: B20180054.
Changsha Science and Technology: kq1706072.

### Competing Interests
The authors declare that they have no competing interests.

### Author Contributions
- Xinyu Li conceived and designed the experiments, performed the experiments, analyzed the data, prepared figures and/or tables, and approved the final draft.
- Yufeng Cai conceived and designed the experiments, performed the experiments, prepared figures and/or tables, and approved the final draft.
- Yinghe Ding analyzed the data, prepared figures and/or tables, and approved the final draft.
- Jia-Da Li analyzed the data, prepared figures and/or tables, and approved the final draft.
- Guoqing Huang analyzed the data, authored or reviewed drafts of the paper, and approved the final draft.
- Ye Liang conceived and designed the experiments, performed the experiments, authored or reviewed drafts of the paper, and approved the final draft.
- Linyong Xu conceived and designed the experiments, authored or reviewed drafts of the paper, and approved the final draft.

### Data Availability
The codes of the current study are available in the GitHub repository, https://github.com/coolleafly/COV_SIM/. The average data generated by 10 simulations for each model are included in the article and the Supplemental Files.

### Supplemental Information
Supplemental information for this article can be found online at http://dx.doi.org/10.7717/peerj.11629#supplemental-information.

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
