# Peer review of "Discrete simulation analysis of COVID-19 and prediction of isolation bed numbers"

_PeerJ, doi:10.7717/peerj.11629_

## Round 0.1 · original submission · Major Revisions

Thank you for submitting your manuscript. Should you choose to revise and resubmit, please pay careful attention to Reviewer #1 and #2's recommendations. Of particular detail, please consider Reviewer #1's discussion specific to research and design.

·

Basic reporting

The research utilized a discrete simulation for modelling of the epidemic spread at the city level. They used the results to derive a simple equation aiming to estimate the number of required isolation beds. Despite the title and general remarks throughout the text, the work is not directly linked to COVID-19 spread.
Authors define the gap in epi-/pandemic modelling, leaving decision-makers with no applicable tool to predict the number of required isolation beds. I greatly appreciate their attempt to develop an applicable tool, as well as generalization of parameters to the level of making said parameters easy to understand.
This work, despite its great value as groundwork and a solid basic research foundation, however, requires modifications in the area of its claimed applicability criteria (explained in detail below).

Experimental design

The modelling and technical efforts are adequate for a general, basic research model which may serve as a base for further research.
The authors shall either: a) align their model results to real-world city-level data, if available to them, and present the use-case scenario or b) clearly present limitations of their theoretical approach and avoid claims suggesting that this method may/shall be used as-is by the non-research community. In the case of taking the former approach, the authors may decide to keep using COVID-19 keyword, if the data used will concern the recent SARS-CoV-2 outbreak. If taking the latter approach, the work shall be presented as a general model (which in this reviewer’s personal opinion does not diminish its value at the slightest).
I would propose re-structuring of the reporting structure to better reflect the goal of the paper which, to my understanding, is the development of the tool usable in the non-research environment. The new structure shall show a clear distinction between two sets of analysed model parameters: 1) biological agent-dependent variables, namely (i) the incubation period and (ii) the healing time; and 2) healthcare system-dependent variables, namely (iii) hospital response time and (iv) the population mobility. If the authors would decide to follow previously mentioned COVID-19 use-case scenario, biological agent-dependent variables shall reflect the ongoing discussion on COVID-19 parameters, and healthcare system-dependent variables shall try to represent the situation in the city used as a research case.

Validity of the findings

For the purposes of the presented research, following ANOVA with pairwise comparisons is an acceptable approach. However, pairwise comparisons shall provide confidence intervals of the results, instead of limiting themselves to providing p-values. The provided confidence intervals shall be also interpreted in the context of the effect size.
The presented findings are sufficient for an initial general theoretical model, however, they lack the connection to real-world data to present them as a practical tool or as reflecting COVID-19 situation.

Additional comments

Minor comments:
Line 33: “We established a discrete simulation model for epidemiology.” - This general statement shall be made more precise by adding two bits of information: 1. It’s a city-level model. 2. It’s derived from an existing modelling framework.
Line 55: If authors decide to use COVID-19 as a use-case scenario, the presented “current status” data shall be updated.
Line 65-66: Definitions of parameters shall be presented as they are first invoked.
Lines 79-80: Definition of states shall be provided as they are first invoked.
Lines 89-90: “The probability follows the random number model and is set by the probability value.” Unclear sentence. Please rephrase.
Line 107: If the authors decide to use real-world data to compare the results of their modelling effort to the real-life COVID-19 example, this assumption shall be discussed in the light of recent findings. (To make the model even more suitable for the application purposes, it would be better to have it parametrized, even if in the presented research the parameter would be set to 0% of the cured population switching to susceptible population, however, if it required more effort than the authors may offer, it’s not a requirement!)
Line 150: The parameter in discussion shall be described properly (not “the hospital cure time”, but “the population mobility”)
Line 199: Please provide the definition of Rt here, as it’s the first time it’s invoked.
Line 274: Please explain in detail what is an assumed definition of “reasonable” in “reasonable number of isolation beds”.
Lines 282-283: Detailed instructions of T1 and T2 estimations shall be presented.
Line 349: See comment on line 274
Lines 360-361: See comment on lines 282-283
Line 375: If the research will not be supplemented with real-life data use-case scenario, COVID-19 related remarks shall be generalized.
Line 396: Please explain what is an assumed definition of “reasonable” in “reasonable request”.

Reviewer 2 ·

Basic reporting

In this study on using computer discrete simulation to analyze the epidemiological features of COVID-19 and provided a possible prediction of isolation bed numbers in response to the pandemic. This study can provide a critical/useful tool for the effectiveness analysis of emerging infectious diseases like COVID-19. However, some critical points modeling and epidemiological aspects should be considered for publication in PeerJ.

There are certain minor language issues that needs attention from the authors.

1. The term "cure time " were used in Lines 36, 40, 43, 139, 148, 369 whilst in the rest of the manuscript, the authors seemed to be using the term "healing time" instead. I wonder if the two terms are actually of the same meaning or they meant differently? If former, then I would suggest the authors unify the terminology to avoid confusion.

2. "represents" would be a better word to replace "means" in Line 276.

3. Lines 317-318, I am not sure if the authors are trying to express the meaning similar of the following "In reality, during the peak of the pandemic, medical staff often have limited access, or no access at all, to personal protective equipment and other medical resources."

4. It is good to reference the Lancet publication describing shelter hospitals practice in China. however, please use the correct name ("Fangcang" with Capital F) to refer the shelter hospitals in Line 340, as this name is a translation from Chinese.

5. Remove the extra "and" in Line 371.

Experimental design

no comment

Validity of the findings

The authors should provide further illustration and critically review regarding the advantages and limitations in the Discussion Part of the study.

---

## Round 0.2 · Minor Revisions

Please review the comments and request made by reviewer number two. I do believe that we will be able to move forward with publication when this reviewers comment/request is addressed.

·

Basic reporting

Despite the title and general remarks throughout the text, the work, in my humble opinion is still not directly linked to COVID-19 spread. That being said, I recognize the value of the described model, and as it describes any "epidemic", COVID-19 can be treated as a subset of potential applications.

Experimental design

The presented structure is not technically incorrect.

Validity of the findings

The presented findings are sufficient for an initial general theoretical model.

Reviewer 2 ·

Basic reporting

I feel appreciated to the author's response and the modifications made to the original manuscript regarding the language issues. The current presentation of your study is now much easier to understand to a broader international audience.

Experimental design

no comment

Validity of the findings

I would very much like to see the authors to actually include the response towards my previous comment of this part, which is "provide further illustration and critically review regarding the advantages and limitations in the Discussion Part of the study". I understand that the authors might have overlooked my one-sentence comment in this part.

I believe that the other reviewer has helped the authors identified certain areas of limitation regarding this study and based on your reply (i.e. to include real-life data etc.), you decided to go further with such area in another publication. I do think this is acceptable given the scope and impact of this potential publication, and your potential next publication might depend on such publication. Nevertheless, I would once again suggest the authors include at least such limitation in the discussion part.

---

## Round 0.3 · Minor Revisions

Dr. Liang - Both reviewers have recommended to accept your manuscript. However in reviewing the manuscript I noted that line 346 notes "Since there is no specific treatment or vaccine for COVID-19, containment of the epidemic depends more on traditional public health measures (Wilder-Smith & Freedman 2020)". While this was the case when you conducted your research, I think the manuscript should be revised to say something on the order of:

"...at the time of the conceptualization of the simulation model..." I do think your process is valid and expands the modeling and simulation knowledge base and thus remains relevant. To this end you may wish to say in the discussion that models such as this one can be adjusted to account for emerging treatments and vaccines (assuming this is an accurate statement in relationship to your model).

I do understand that you have been working hard and diligently on this manuscript for some time, but also want your work to be relevant to current conditions. Thank you for your patience during these rapidly evolving times.

Reviewer 2 ·

Basic reporting

no comment

Experimental design

no comment

Validity of the findings

The authors had made a comprehensive summary regarding the advantages and limitations towards the study they described in the manuscript, which makes the manuscript more critically illustrated. During the process, the authors had helped to improve the manuscript as the current contents now provide more insights for further research and investigation.

Additional comments

I encourage the author team to continue with possible further research directions listed in the manuscript and see more joint research in computer science and healthcare management, which I foresee the potential of discrete simulation in healthcare in China.

---

## Round 0.4 · accepted · Accept

Thank you for your perseverance and patience as it relates to this manuscript. As noted previously preparing a manuscript for publication is often a very iterative process. I am delighted to see this manuscript moving forward.

Reviewer 2 ·

Basic reporting

no comment

Experimental design

no comment

Validity of the findings

no comment